# How charitable giving affects litigation duration? Empirical evidence from China

**Feng Zhu** [ORCID] *

School of Law, Southwestern University of Finance and Economics, Chengdu, China

* 1220201z1001@smail.swufe.edu.cn

**Data Availability Statement:** The data that support the findings of this study are openly available in Data and procedures at http://dx.doi.org/10.5281/zenodo.14238241.

**Funding:** The author(s) received no specific funding for this work.

## Abstract

Charitable donations are an important manifestation of corporate social responsibility. Current research focuses on the economic effects of corporate donations while ignoring their legal effects in the litigation field. This paper utilizes litigation and arbitration data from A-share listed companies in Shanghai and Shenzhen from 2008 to 2021 to investigate the impact and mechanism of charitable donations on the litigation duration of listed companies. The study finds that: (1) Charitable donation behavior can significantly shorten the litigation duration of listed companies. (2) This effect is particularly significant when listed companies are plaintiffs, but not significant when they are defendants. (3) The negative relationship between charitable donations of listed companies and litigation duration is achieved through reputation mechanisms. (4) This effect is more significant for listed companies located in the Northeast region and regions with a lower degree of marketization when facing civil and arbitration cases. (5) Additionally, by constructing panel data on whether companies are involved in litigation and as defendants, it is shown that corporate charitable donations not only have the function of "compensating" reputation but also serve as a kind of "insurance" for reputation. After robustness tests and overcoming endogeneity issues, the above conclusions still hold. This paper clarifies the implicit interaction mechanism between listed companies and judicial departments, enriches research in the field of organizational reputation theory, and is of significant importance for deeply understanding the motivation of corporate charitable donations.

## Introduction

Corporate reputation represents a significant intangible asset that profoundly influences consumer behavior, market direction, and societal opinion [1]. The events surrounding Wahaha in 2024 underscore the substantial impact corporate reputation can have on stakeholders such as consumers, competitors, and investors. Consequently, numerous enterprises, particularly listed companies, are augmenting their investments in reputation assets, with charitable donations serving as a pivotal strategy [2].

The "China Corporate Philanthropy Development Report (2023)" published by the China Social Security Society reveals that Chinese corporate philanthropic donations have reached

**Competing interests:** The authors have declared that no competing interests exist.

hundreds of billions of dollars, comprising approximately 60% of the country's total statistically significant donations. This indicates that corporate donations have become a fundamental pillar of philanthropy in China, positively contributing to the goal of common prosperity. However, explaining the obsession of listed companies with philanthropy solely through the fulfillment of social responsibility appears inadequate. Extant literature has extensively delved into the argument that the altruistic nature of listed companies' charitable donations brings additional benefits to the enterprises, thereby rendering the motivation for such donations multifaceted [3]. Enterprises that actively engage in charitable giving are more likely to gain the goodwill and trust of local governments [4]. This goodwill and trust can serve as a favorable tool for establishing political relations and protecting corporate property rights [5]. Subsequently, such engagement can facilitate access to financing facilities, government subsidies, and investment opportunities for the enterprises [6].

The motives for charitable giving among public companies are multifaceted, encompassing not only the fulfillment of social responsibility but also strategic considerations such as generating additional revenue, fostering political relationships, and delivering various benefits to the company. However, the role of charitable giving in litigation remains underexplored. Inevitably, listed companies become involved in litigation, either actively or passively, during their operational processes. In the short term, litigation serves as a red flag indicating a corporate crisis, while in the long term, it can significantly impact a company's reputation, financial standing, and future prospects. Given their status as public interest entities, listed companies must prioritize managing litigation. Dai et al. [7] argue that charitable donations can be seen as a self-redemptive mechanism for companies facing high litigation risks, suggesting that listed companies may utilize charitable giving as a strategic public relations measure. Fu et al. [8] empirically demonstrate that litigation risks faced by Chinese listed companies can significantly elevate the level of charitable donations, grounded in reputational insurance theory. Existing research primarily focuses on the overall risk control level of litigation events, highlighting the influence of corporate donations on litigation risk, but lacks a thorough analysis of the potential effect of charitable donations on the likelihood of litigation. Specifically, it remains unclear whether corporate charitable donations could function as a means for companies to acquire undisclosed benefits during litigation processes. Current research on the interplay between charitable donations and litigation cycles of listed companies is sparse, lacking a comprehensive theoretical framework and empirical analysis. Therefore, this paper aims to provide theoretical insights and practical guidance for listed companies in fulfilling their social responsibilities and managing litigation risks by investigating the relationship between charitable donations and litigation cycles, as well as exploring the underlying interaction mechanisms and influencing factors.

The marginal contributions of this paper are fourfold. Firstly, it broadens the research scope of corporate charitable donations to the realm of judicial litigation, empirically examining the actual impact of such donations on the litigation cycle of listed companies [8]. This not only augments the research on the economic consequences of corporate charitable donations, but also offers a novel perspective on the interplay between CSR and the judicial environment. Secondly, the paper sheds light on the specific mechanism of charitable donations' influence on the litigation cycle of listed companies, considering varying litigation statuses. Thirdly, it injects fresh impetus into the study of external influences in judicial litigation, grounded in reputation theory. By elucidating how charitable donations alter the case processing order and resource allocation of judicial activities through their effect on corporate reputation, this paper provides a novel theoretical underpinning for comprehending the complexity and externalities of judicial activities, thereby enriching the cross-disciplinary study of jurisprudence and economics. Finally, the paper accounts for variations in the judicial effects of charitable donations

across different contexts, including regional disparities and case type distinctions. This analysis not only bolsters the study's comprehensiveness and depth, but also equips policymakers with more nuanced policy recommendations to steer corporate charitable giving behavior and foster societal harmony.

The structure of the aforementioned paper is delineated as follows: In Part II, the theoretical underpinnings and hypotheses are examined. Part III delves into the empirical methodology and presents pertinent sample statistics. Part IV offers a detailed analysis of the empirical findings. Subsequently, Part V conducts a robustness test to validate the results. Part VI provides an in-depth exploration of the underlying mechanisms, while Part VII undertakes further exploration. Finally, the paper concludes with a summary of the key findings and insights gained.

## Literature review and research hypothesis

The term "charitable donations" inherently implies voluntariness and non-compensation. A substantial body of literature supports the view that corporate charitable donations are an expression of social responsibility [9–14]. However, as research and practice have advanced, it has become apparent that charitable donations, typically seen as altruistic acts, can also yield additional benefits for firms, thus complicating the motivation behind such donations [3, 15–17]. Firms, as rational economic entities, are driven by the pursuit of economic benefits. If charitable donations can generate above-average returns, more firms are likely to engage in such philanthropic activities. Literature on self-interested corporate charitable donations indicates that firms can derive economic benefits [6, 18, 19] and political advantages [4, 20–23] from these activities. Both domestic and international research has extensively highlighted that charitable donations not only provide direct benefits to companies, such as enhanced access to financing, government subsidies, and investment opportunities, but also offer indirect advantages in areas such as politics [5, 24], markets [15], and the judiciary. Nguyen and Phan [25] suggest that one of the reasons companies engage in corporate social responsibility (CSR) activities is to reduce shareholder litigation risks and mitigate the consequences of post-litigation settlements. Furthermore, both pre-litigation risks and post-settlement costs can further influence a company's credit rating and credit quality [26]. Research by Fu et al. [8] indicates that charitable donations can act as a form of reputation insurance for companies facing litigation risks. As a crisis buffer, charitable donations can protect corporate value, market reactions, and stock prices from excessive negative impacts [27–29]. However, litigation is a complex legal process that involves not only preemptive risk assessment and post-judgment outcomes, but also the often-overlooked litigation process itself. Existing research on the relationship between charitable donations and judicial activities has primarily focused on pre-litigation risks and post-litigation outcomes [30–32], while neglecting the study of the litigation process itself.

To clarify whether charitable donations can play a substantive role in the actual litigation process of listed companies, two logical challenges must be addressed: First, the judiciary is relatively independent; how can it be influenced by corporate charitable donations? Second, the litigation cycle is explicitly regulated by relevant legal provisions; how can it vary across different companies?

To address the first logical challenge, we use Reputation Theory to provide an explanation [33]. Research by Ng and He [34] indicates that law is not the sole criterion for judicial decisions in China; political, administrative, social, and economic factors all influence verdicts. As an important manifestation of corporate social responsibility, charitable donations are also a significant factor influencing judicial activities [25]. Existing research explains how political

mechanisms, social network mechanisms, and reputation mechanisms influence judicial activities through corporate social responsibility behaviors, such as charitable donations. Lu et al. [35] found that Chinese courts show a bias in favor of state-owned enterprises and private enterprises with personal political connections in commercial litigation. Wang and Qian [36] also noted that corporate involvement in charitable activities helps firms gain social and political legitimacy, which in turn has a positive impact on their financial performance. Corporate charitable donation behavior, as a social phenomenon, although not directly linked to the core functions and decision-making processes of judicial departments, can indirectly influence judicial decisions and actions through various channels. Since the 19th century, economists such as Adam Smith have focused on the impact of reputation on individual behavior and corporate operations. Reputation theory emphasizes that reputation is the public's overall evaluation of an individual or organization, formed based on past behaviors and performance within a specific social context [37]. Reputation is an intangible asset that significantly influences economic behavior, requiring joint efforts from both businesses and society to establish and maintain. The reputation that a company forms acts as a signaling mechanism [38] to relevant stakeholders, thereby influencing their evaluations and actions. Field et al. [39] empirically demonstrate that companies disclosing favorable information can deter certain types of lawsuits. Cahan et al. [40] found that corporate social responsibility actions can influence media behavior by enhancing media goodwill toward the company. Based on reputation theory and signaling theory, it can be inferred that judicial authorities are not isolated from relevant information. Judicial bodies receive reputation signals emitted by litigants, which in turn influence their actions. Corporate charitable donations can enhance a firm's social image and public reputation, thereby affecting the behavior of judicial authorities. Judicial personnel may take into account a company's performance in fulfilling social responsibilities as a reference when exercising discretionary power. Discussions on the social network mechanism often focus on corporate executives as research subjects, which will not be addressed in this study.

To address the second issue, it is necessary to carefully examine how China's litigation legal system is structured. According to Articles 164 and 152 of the Civil Procedure Law, cases that fall under the simplified or ordinary procedures are required to be resolved within a first-instance period of "3+1+x" and "6+6+x" months, respectively. This means that, under normal circumstances, a civil lawsuit should be concluded within four months for simplified cases and within twelve months for ordinary cases. In the context of a high caseload and fewer judges, China's litigation law establishes a maximum trial duration but does not set a minimum timeframe, thereby providing judicial authorities with some flexibility in determining the processing order and timeline for cases. The similar provisions for administrative, criminal, and arbitration cases (the relevant legal texts are listed in the S1 Appendix) illustrate that judicial authorities have discretionary power over the litigation duration involving companies. Further examination of the provisions in the Civil Procedure Law reveals that the litigation cycle of a case can exceed the aforementioned maximum duration for two reasons: first, due to "x," the time extension after approval by a higher-level people's court varies; second, according to the relevant provisions of the Civil Procedure Law, time spent on jurisdictional objections, evaluations, appraisals, audits, suspension of proceedings, and other factors is not included in the trial time limit. It can be concluded that the theoretical range for the litigation cycle is any positive integer, and the duration of the litigation cycle is subject to the discretion of the judiciary. Therefore, it is evident that China's judicial authorities, in handling lawsuits involving listed companies, operate under conditions where they may be influenced by factors beyond the law (such as corporate charitable donations), and such influences may affect the litigation cycle at the procedural law level.

The process of litigation for listed companies not only consumes financial resources [31], but also depletes significant time resources. More importantly, it may lead to a decline in the company's reputation value, ultimately jeopardizing its long-term development [41]. Therefore, as a listed company, there is both the motivation and the drive to seek "preferential treatment" during the litigation process: to shorten the litigation cycle as much as possible, reduce litigation costs, and minimize reputation loss. For the judiciary, it is impossible to fully understand all the information from both parties (the plaintiff and defendant), so they can only form initial impressions based on limited public information. According to Bayesian theory, when faced with incomplete information, individuals tend to rely on the frequency of events related to the specific nature of the matter to judge the probability of its inherent attributes. Specifically, in the litigation context, decision-makers form subjective impressions of a company based on its public information. This means that if a company is known for its charitable actions, it is often perceived as a good company. The positive judgment formed by decision-makers can be seen as the realization of the reputation insurance mechanism mentioned in previous studies [8], which allows the company to incur lower litigation costs. Based on this, we propose Hypothesis 1:

H1: Corporate charitable donations have a significant negative impact on the litigation cycle of listed companies.

When a listed company acts as either the plaintiff or defendant, its subjective attitude differs. As the plaintiff, the company is typically the party asserting its rights, seeking and hoping to obtain judicial support as quickly as possible in order to achieve its claims. In such cases, the company is more likely to aim to shorten the litigation cycle. If the reputational effect of corporate charitable donations can function as a form of insurance, then when a company engaging in charitable donations acts as the plaintiff, the litigation cycle may be shorter. Thus, Hypothesis 2 is proposed:

H2: When a listed company is the plaintiff, the negative impact of charitable donations on the company's litigation cycle is greater.

In contrast, when a company is the defendant, it is typically the party whose rights are being contested. Once a litigation judgment is issued, it is likely to result in the outflow of the company's resources. From this perspective, compared to when the company is the plaintiff, the company may prefer a longer litigation period. However, as the previous analysis has indicated, litigation itself imposes additional costs on the company, including human, material, financial resources, and intangible assets such as reputation. Therefore, when a listed company is the defendant, there are two opposing forces at play, creating uncertainty in the relationship between charitable donations and the litigation cycle. Thus, Hypothesis 3 is proposed:

H3: When a listed company is the defendant, the negative impact of charitable donations on the company's litigation cycle is smaller or not significant.

## Research design

### (i) Sources of data and sample selection

This paper utilizes litigation and arbitration data for Shanghai and Shenzhen A-share listed companies spanning from 2008 to 2021. The initial sample comprises 3756 observations, where both the prosecution date and the first-instance judgment date are recorded. Samples were then screened based on the following criteria: (1) exclusion of financial industry firms, (2) removal of ST-classified samples, (3) elimination of samples with missing donation data,

(4) removal of anomalous samples with litigation periods less than or equal to 0, and (5) elimination of duplicate case records. The resulting mixed cross-sectional dataset contains 2990 observations across 811 listed companies. Litigation data for listed companies were sourced from the CNRDS database, while charitable donation data was derived from the non-operating income or expenditure disclosed in the CSMAR database. The annual donation amount of listed companies was calculated by manually consolidating all donation items, serving as the foundation for constructing two key explanatory variables: the binary variable indicating whether a donation was made and the donation scale. All other control variable data were obtained from the CSMAR database. To mitigate the impact of outliers on the study's results, a 1% Winsorize treatment was applied to all continuous variables.

## (ii) Modeling and variable setting

To investigate the impact of charitable donations on the litigation cycle, this study formulates the following regression model:

$$\text{Time}_{it} = \beta_0 + \beta_1\text{Donate}_{it} + \delta\text{Controls}_{it} + \varepsilon_{it}$$

The litigation cycle, denoted as $\text{Time}_{it}$, serves as the explanatory variable in this study. It is defined as the ratio of the number of working days between the indictment date and the date of the first instance judgment to the total number of legal working days in a year (250 days). During the data preparation phase, it was observed that 48 outlier samples existed, characterized by the first trial verdict date preceding the indictment date. These outliers were excluded from further analysis. The core explanatory variable in this paper is charitable giving of listed companies, denoted as $\text{Donate}_{it,}$ which is measured both in terms of the presence or absence of donations and the logarithm of the donated amount.

This paper incorporates control variables pertaining to firm financial characteristics, corporate governance, and regional characteristics. Specifically, to account for firm financial characteristics, we employ the natural logarithm of total assets at the end of the period (Size) to capture the impact of firm size, relative cash holdings (Cash) as a proxy for redundant resources, and control for ROA and Leverage. To address corporate governance, we follow the study of Lee and O'Neill [42] and introduce the degree of equity concentration (Shrcr1), the degree of separation of powers (Separation), the nature of corporate property rights (Soe), and the growth potential (Growth) as control variables. As the subject of this paper is a listed company, the environment in which it is located will be affected by the year, the region and the industry, specifically the litigation cycle studied in this paper, the listed company itself is located in the industry has less influence, while the year and the region have a greater impact, so the year and the province have been controlled to a certain extent. In order to control for regional characteristics, two control variables are set for the government-market relationship (Government) and the development of market intermediary organizations and the legal system environment (Law) in the province where the enterprise is registered. Year fixed effects are also included to control for year characteristics. The precise definitions of these variables are outlined in Table 1.

## (iii) Descriptive statistics and univariate analysis

The descriptive statistics of the key variables are presented in Table 2. Specifically, the litigation cycle's mean value of 1.009 suggests that the average litigation duration for listed companies approximates one year. The mean value of the donation variable (D_donate) stands at 0.616, indicating that approximately 62% of the sampled enterprises engage in charitable donations. This aligns with the donation data released by prominent domestic organizations, thereby

**Table 1. Variable definitions.**

| Dependent variables | |
|---|---|
| Time | Number of working days between the date of indictment and the date of first instance judgment / the total number of legal working days in a year |
| **Independent variable** | |
| Lndonate | Listed companies annual donations plus 1 to take the logarithm of the amount obtained |
| D_donate | If the listed company's annual donation amount is greater than 0, then assigned a value of 1, otherwise 0 |
| **Control variables** | |
| Size | Logarithmic value of total assets |
| Cash | Money funds/total assets |
| ROA | Net profit/total assets |
| Leverage | (Net profit + income tax expense + financial expense) / (net profit + income tax expense) |
| Shrcr1 | Shareholding ratio of the company's first largest outstanding shareholders |
| Seperation | Difference between control and ownership of listed companies by beneficial owners |
| Soe | By state-owned enterprises and non-state-owned enterprises |
| Growth | (Total assets at the end of the current period—total assets at the end of the previous period)/ total assets at the end of the previous period |
| Law | Development of Market Intermediary Organizations and Legal System Environment Provincial Indicators |
| Government | Provincial Indicators of Government-Market Relationship |

validating the representativeness of the research data. Furthermore, the standard deviation of the enterprise donation scale (Lndonate) reveals a significant disparity in donation amounts among listed companies.

Among the control variables, the mean value of the net profit margin of total assets (ROA) is-0.053, indicating generally poor business performance for litigation-involved enterprises. This concurs with the notion that litigation serves as an indicator of corporate crisis, thus justifying the rationale for managing the litigation cycle of these enterprises.

Table 3 reports the findings of the univariate between-group mean difference analysis. This analysis was conducted to examine the association between litigation status, donation status,

**Table 2. Descriptive statistics.**

| Variables | Obs. | Mean | SD | Minimum | Maximum |
|---|---|---|---|---|---|
| Time | 2,990 | 1.009 | 1.083 | 0.004 | 9.684 |
| D_donate | 2,990 | 0.616 | 0.486 | 0 | 1 |
| Lndonate | 2,990 | 7.394 | 6.156 | 0 | 20.062 |
| Soe | 2,773 | 0.559 | 0.497 | 0 | 1 |
| Leverage | 2,168 | 1.685 | 1.420 | 0.533 | 9.554 |
| Shrcr1 | 2,774 | 18.856 | 17.895 | 0.116 | 64.143 |
| Size | 2,789 | 21.931 | 1.321 | 18.951 | 25.178 |
| Growth | 2,937 | 0.100 | 0.480 | -0.703 | 3.431 |
| ROA | 2,968 | -0.053 | 0.281 | -1.994 | 0.248 |
| Seperation | 2,447 | 4.816 | 7.497 | 0 | 27.090 |
| Cash | 2,946 | 0.139 | 0.129 | 0.001 | 0.678 |
| Law | 2,990 | 8.681 | 3.120 | 1.800 | 15.189 |
| Government | 2,990 | 8.043 | 1.436 | 4.150 | 11.175 |

**Table 3. Differences in between-group means of time.**

| | By amount donated or not | | By median amount donated | |
|---|---|---|---|---|
| | YES | NO | HIGH | LOW |
| Company as Plaintiff | 0.936 | 1.026 | 0.903 | 1.035 |
| Difference in mean values | 0.091* | | 0.131*** | |
| Company as Defendant | 1.164 | 0.945 | 1.189 | 0.967 |
| Difference in mean values | -0.218*** | | -0.222*** | |

Note: ***, ** and * indicate significance at 1%, 5% and 10% level of significance, respectively, as in the table below.

and litigation cycle. When firms serving as plaintiffs in lawsuits are considered, those that donate tend to have shorter litigation cycles compared to non-donating firms. Moreover, within the donating firms, those with higher donation amounts exhibit shorter litigation cycles compared to firms with lower donation levels. Conversely, when firms are defendants in lawsuits, the pattern reverses, with donating firms experiencing longer litigation cycles than non-donating firms. Similarly, among donating firms, those with higher donation amounts have longer litigation cycles than firms with lower donation levels. While these findings align broadly with the theoretical analysis, further regression analysis is necessary to substantiate these observations.

## Empirical results and analysis

The litigation cycle, as a continuous variable, is analyzed using an ordinary least squares (OLS) regression model. Table 4 summarizes the key estimation results pertaining to Hypotheses 1–3. Columns (1) to (4) specifically focus on Hypothesis 1, examining the impact of corporate donations on litigation cycles. Columns (1) and (2) analyze whether a firm donates (D_donate) as an explanatory variable, with the inclusion of year fixed effects in a step-wise manner. The results indicate that, at the 10% significance level, a firm's decision to donate is negatively correlated with its litigation cycle, suggesting that charitable giving behavior can effectively reduce the duration of legal disputes. Columns (3) and (4) further explore the scale of donations (Lndonate) as an explanatory variable. The regression outcomes reveal a significant negative correlation between donation scale and litigation cycle at the 10% significance level, indicating that an increase in donation scale leads to a shorter litigation cycle, thereby supporting Hypothesis 1.

Columns (5) and (6) present the results testing Hypothesis 2. The regression analysis indicates that, for listed companies as plaintiffs, their litigation cycle negatively correlates with both the likelihood of donating and the donation amount at a 1% significance level. Furthermore, a comparison of regression coefficients across columns (1) to (4) reveals that, compared to the entire sample, the negative influence of charitable donations on the litigation cycle is more pronounced when the listed company is the plaintiff. These empirical findings support Hypothesis 2.

Columns (7) and (8) present the test results for Hypothesis 3. The regression analysis indicates that, for listed companies as defendants, the relationship between the litigation cycle and the likelihood of donating as well as the donation amount is not significant. This is attributed to the company's dual concerns: avoiding the unfavorable signal of prolonged litigation involvement and preventing a rapid resolution that could result in the loss of real economic benefits. These opposing forces balance each other, rendering the coefficient insignificant. The empirical findings thus support Hypothesis 3.

**Table 4. Key estimation results for scenarios 1–3.**

| Variables | (1) | (2) | (3) | (4) | (5) | (6) | (7) | (8) |
|---|---|---|---|---|---|---|---|---|
| | Full Sample | | | | Company as Plaintiff | | Company as Defendant | |
| D_donate | -0.124* | -0.120* | | | -0.201*** | | -0.002 | |
| | (0.064) | (0.064) | | | (0.072) | | (0.138) | |
| Lndonate | | | -0.010** | -0.009* | | -0.016*** | | 0.00200 |
| | | | (0.005) | (0.005) | | (0.006) | | (0.0107) |
| Soe | -0.024 | -0.061 | -0.027 | -0.063 | -0.010 | -0.017 | -0.022 | -0.0184 |
| | (0.060) | (0.059) | (0.060) | (0.059) | (0.073) | (0.072) | (0.115) | (0.114) |
| Leverage | -0.019 | -0.025 | -0.019 | -0.024 | -0.020 | -0.021 | -0.039 | -0.0381 |
| | (0.021) | (0.022) | (0.021) | (0.022) | (0.025) | (0.025) | (0.033) | (0.0320) |
| Shrcr1 | 0.002 | 0.003* | 0.002 | 0.003* | -0.001 | -0.001 | 0.006** | 0.00609** |
| | (0.001) | (0.001) | (0.001) | (0.001) | (0.002) | (0.002) | (0.003) | (0.00308) |
| Size | 0.133*** | 0.156*** | 0.140*** | 0.161*** | 0.190*** | 0.200*** | 0.109** | 0.104** |
| | (0.025) | (0.026) | (0.027) | (0.028) | (0.029) | (0.030) | (0.048) | (0.0522) |
| Growth | 0.079* | 0.021 | 0.078* | 0.020 | -0.170** | -0.171** | 0.143* | 0.141* |
| | (0.047) | (0.050) | (0.047) | (0.050) | (0.075) | (0.075) | (0.082) | (0.0816) |
| ROA | 0.635 | 0.270 | 0.677 | 0.310 | 0.226 | 0.243 | 0.219 | 0.219 |
| | (0.670) | (0.611) | (0.671) | (0.610) | (0.815) | (0.817) | (0.918) | (0.906) |
| Seperation | -0.008** | -0.008** | -0.008** | -0.008** | -0.007* | -0.007* | -0.011 | -0.0109 |
| | (0.003) | (0.004) | (0.003) | (0.004) | (0.004) | (0.004) | (0.007) | (0.00714) |
| Cash | -0.179 | -0.111 | -0.173 | -0.104 | 0.343 | 0.337 | 0.014 | 0.00844 |
| | (0.213) | (0.211) | (0.212) | (0.210) | (0.270) | (0.270) | (0.356) | (0.356) |
| Law | -0.052*** | 0.000 | -0.052*** | -0.000 | -0.035** | -0.036** | 0.047 | 0.0461 |
| | (0.009) | (0.014) | (0.009) | (0.014) | (0.018) | (0.018) | (0.029) | (0.0288) |
| Government | 0.140*** | 0.026 | 0.140*** | 0.026 | 0.108*** | 0.108*** | -0.063 | -0.0612 |
| | (0.021) | (0.027) | (0.021) | (0.027) | (0.037) | (0.037) | (0.050) | (0.0495) |
| Constant | -2.486*** | -2.548*** | -2.630*** | -2.673*** | -3.968*** | -4.192*** | -0.876 | -0.798 |
| | (0.579) | (0.588) | (0.608) | (0.618) | (0.680) | (0.710) | (1.038) | (1.110) |
| Year | NO | YES | NO | YES | YES | YES | YES | YES |
| Observations | 1,710 | 1,710 | 1,710 | 1,710 | 1,010 | 1,010 | 633 | 633 |

Notes:1. Z-statistics in parentheses, same as in the table below.

2.The standard errors are heteroskedasticity robust standard errors.

3. The discrepancy of 67 observations between the subgroup sums and the total sample is attributable to the presence of 'third-party' involvement and 'uncertainty' in the litigation status of listed companies.

The regression results in Table 4 are also noteworthy for the coefficients related to the development of market intermediary organizations, the legal institutional environment (Law), and the government-market relationship (Government). Based on the results of columns (5) and (6) of Table 4, it is evident that in the regression analysis of listed companies as plaintiffs, the regression coefficients of the former are significantly negative, while those of the latter are significantly positive. This suggests that a more developed legal system environment and market intermediary organizations lead to a shorter litigation cycle, indicating higher litigation efficiency. This, to some extent, reflects the positive impact of accelerating the construction of a society based on the rule of law and a community of legal professions. Conversely, a closer relationship between the government and the market results in a longer litigation cycle, affirming the necessity for China to advance market-oriented reforms, strengthen regulatory mechanisms, and establish a mature mechanism for separating government and enterprises.

**Table 5. Robustness tests.**

| Variables | (1) | (2) | (3) | (4) | (5) | (6) | (7) | (8) |
|---|---|---|---|---|---|---|---|---|
| | Full Sample | | | | Company as Plaintiff | | Company as Defendant | |
| D_donate | -0.119* | -0.115* | | | -0.193*** | | -0.002 | |
| | (0.061) | (0.062) | | | (0.069) | | (0.133) | |
| Lndonate | | | -0.009** | -0.009* | | -0.015*** | | 0.002 |
| | | | (0.005) | (0.005) | | (0.005) | | (0.010) |
| Constant | -2.388*** | -2.447*** | -2.525*** | -2.567*** | -3.811*** | -4.025*** | -0.843 | -0.768 |
| | (0.555) | (0.564) | (0.583) | (0.593) | (0.651) | (0.680) | (0.996) | (1.065) |
| Controls | YES | YES | YES | YES | YES | YES | YES | YES |
| Year | NO | YES | NO | YES | YES | YES | YES | YES |
| Observations | 1,710 | 1,710 | 1,710 | 1,710 | 1,010 | 1,010 | 633 | 633 |

## Robustness tests

### (i) Remeasurement of the dependent variable

In the benchmark regression analysis, this paper quantifies the litigation cycle by calculating the ratio of the number of legal working days from the initial prosecution to the final resolution of the case, divided by the total number of legal working days in a year. To enhance the robustness of our findings, we also employ an alternative measure of the litigation cycle, which is calculated by dividing the number of natural days between the initial prosecution and case conclusion by the total number of days in a year (365 days). For both measures, the same econometric model as in the benchmark regression is utilized for fitting. The results of these regressions are presented in Table 5 and are in line with those obtained from the benchmark regression.

### (ii) Replacement of the estimated model

Given the explanatory variable, litigation cycle, takes on a range of non-negative integers and is a restricted variable, the use of least squares estimation may not accurately capture its true distribution, potentially resulting in estimation bias. Consequently, this study employed the Tobit model for re-estimation to better reflect the actual range of values of the litigation cycle and its potential influencing factors. After implementing this alternative estimation method, the regression results remained consistent with those of the benchmark regression, suggesting the robustness of the original findings. The specific regression results are omitted from the main text but are available for reference.

### (iii) Placebo test

To accurately assess the impact of corporate charitable giving on litigation cycle and mitigate estimation bias from omitted variables, this study adopts a placebo test, following the methodology of Li et al. [43]. Specifically, we randomly assign the values of "D_donate" and "Lndonate" across enterprises, ensuring that these randomly assigned indicators do not accurately reflect the actual donation levels. We then perform regression analysis on these randomly assigned samples. If a significant negative correlation persists between corporate philanthropy and litigation cycle, it suggests a spurious correlation. Conversely, a lack of significance indicates a genuine impact of corporate charitable giving on litigation activities. We repeat this process 100 times, randomly assigning the corporate charitable giving indicator and conducting regressions accordingly. In the majority of these random regressions, the coefficients

associated with charitable giving are insignificant, indicating that our findings are not attributable to pseudo-regression. However, due to space limitations, we do not report the detailed results of the placebo test in this paper.

### (iv) Endogenous issues

According to Fu et al. [8], charitable giving may be influenced by a firm's litigation risk in the preceding period, implying a potential bidirectional relationship between charitable donations and litigation cycles. Furthermore, due to the inability to fully account for factors affecting litigation timelines, omitted variable bias may occur. To mitigate the potential endogeneity issue, which could undermine the robustness of estimation results, this study employs the instrumental variable of average donations by province, industry, and enterprise nature as the principal explanatory variable for a two-stage least squares (2SLS) regression. The "average donation by province, industry, and enterprise nature" serves as a macro-level variable that captures variations in economic development, cultural background, policy environment, industry characteristics, and enterprise nature across provinces. These factors may influence the donation behavior of listed companies, resulting in higher donation levels in economically developed provinces, specific industries (such as public utilities, pharmaceuticals, and biotechnology), and state-owned enterprises. Importantly, since this variable is not directly related to the specific circumstances of individual listed companies and lacks mutual causality, it satisfies the requirements of relevance and exogeneity, qualifying it as an instrumental variable for charitable donations by listed companies.

The regression outcomes in the second stage, presented in Table 6, indicate that the baseline regression results remain valid following the instrumental variable estimation. This suggests that after addressing the endogeneity concern, the conclusions retain their robustness.

### Mechanism analysis

The reputation mechanism may serve as a catalyst for reducing the litigation cycle through charitable donations. Drawing inspiration from the study conducted by Guan and Zhang [1], this research selects 12 core corporate reputation evaluation indicators to establish the corporate reputation rating (Level) variable. Employing the analytical approach outlined by Chen et al. [44], an empirical analysis is undertaken to investigate how corporate charitable donations influence corporate reputation, ultimately affecting the duration of corporate litigation cycles. Given that corporate reputation ratings range from 1 to 10, this study utilizes the

**Table 6. Instrumental variable method.**

| Variables | (1) | (2) | (3) | (4) | (5) | (6) |
|---|---|---|---|---|---|---|
| | Full Sample | | Company as Plaintiff | | Company as Defendant | |
| | Time | Time | Time | Time | Time | Time |
| D_donate | -0.170** | | -0.265*** | | -0.001 | |
| | (0.078) | | (0.091) | | (0.146) | |
| Lndonate | | -0.011* | | -0.020*** | | 0.007 |
| | | (0.006) | | (0.007) | | (0.012) |
| Constant | -2.632*** | -2.745*** | -4.056*** | -4.333*** | -0.874 | -0.616 |
| | (0.575) | (0.605) | (0.678) | (0.716) | (0.983) | (1.040) |
| Controls | YES | YES | YES | YES | YES | YES |
| Year | YES | YES | YES | YES | YES | YES |
| Observations | 1,710 | 1,710 | 1,010 | 1,010 | 633 | 633 |

**Table 7. Validation of reputation mechanisms.**

| Variables | (1) | (2) | (3) | (4) | (5) | (6) | (7) | (8) |
|---|---|---|---|---|---|---|---|---|
| | Level | Level | Level | Level | Level | Level | Time | Time1 |
| D_donate | 0.277*** | 0.055** | 0.049** | | | | | |
| | (0.030) | (0.022) | (0.021 | | | | | |
| Lndonate | | | | 0.030*** | 0.003 | 0.002 | | |
| | | | | (0.002) | (0.002) | (0.002) | | |
| Level | | | | | | | -0.032* | -0.030* |
| | | | | | | | (0.016) | (0.016) |
| Constant | 1.456*** | -6.055*** | -5.877*** | 1.379*** | -6.038*** | -5.876*** | -3.540*** | -3.397*** |
| | (0.027) | (0.173) | (0.180) | (0.028) | (0.181) | (0.188) | (0.848) | (0.813) |
| Controls | NO | YES | YES | NO | YES | YES | YES | YES |
| Year | NO | NO | YES | NO | NO | YES | YES | YES |
| Observations | 1,882 | 1,634 | 1,634 | 1,882 | 1,634 | 1,634 | 1,634 | 1,634 |

Poisson regression method to more precisely capture the relationship between charitable donations and reputation, as the traditional linear regression model may not be fully suited for this purpose. In our further examination of the relationship between public company reputation and the litigation cycle, we employed the same Ordinary Least Squares (OLS) model as the baseline regression.

Table 7 presents the regression results for the relationships between public company charitable giving and reputation, as well as reputation and the litigation cycle. Based on the results from Columns (1) to (6) of Table 7, we found that corporate charitable giving behavior has a significant positive impact on corporate reputation. This suggests that by actively engaging in charitable giving activities, corporations demonstrate their sense of social responsibility and effectively enhance their reputation ratings. Columns (7) and (8) of Table 7 show that corporate reputation significantly shortens the litigation cycle of listed companies, indicating that a good corporate reputation not only brings positive social evaluation and consumer recognition to enterprises but also helps them obtain more favorable treatment when facing legal disputes, such as litigation. In conclusion, charitable donations are an effective way for enterprises to establish a good reputation, which can bring numerous hidden benefits, including favorable treatment in litigation.

## Heterogeneity analysis

### (i) Charitable donations, types of cases and litigation cycles

The diverse nature of litigation cases involving enterprises inherently leads to variations in applicable legal procedures, enterprise attitudes, and judicial discretion. To delve deeper into how case type impacts the relationship between charitable giving and litigation duration, this study conducts a grouped analysis. Based on the litigation and arbitration data of listed companies, the sample is categorized into three groups: civil cases, arbitration cases, and administrative/criminal cases. It is worth noting that within the same case type, courts face identical maximum trial durations. Our analysis focuses on differences in litigation durations among listed companies, and the variations in maximum trial durations across case types do not affect our empirical results or the subsequent analysis of conclusions. As presented in Table 8, the group regression results indicate that charitable donations by listed companies significantly shorten litigation cycles in civil and arbitration cases, yet show no significant effect in administrative and criminal cases. This finding implies that charitable donations of listed companies

**Table 8. Charitable donations, types of cases and litigation cycles.**

| Variables | (1) | (2) | (3) | (4) | (5) | (6) |
|---|---|---|---|---|---|---|
| | Civil Cases | | Arbitration Cases | | Administrative and Criminal Cases | |
| D_donate | -0.130* | | -0.341* | | 0.239 | |
| | (0.069) | | (0.198) | | (0.567) | |
| Lndonate | | -0.010* | | -0.032** | | 0.0183 |
| | | (0.005) | | (0.015) | | (0.049) |
| Constant | -2.491*** | -2.634*** | -5.502*** | -6.151*** | 3.552 | 3.777 |
| | (0.623) | (0.655) | (2.004) | (2.094) | (6.053) | (6.104) |
| Controls | YES | YES | YES | YES | YES | YES |
| Year | YES | YES | YES | YES | YES | YES |
| Observations | 1,565 | 1,565 | 107 | 107 | 38 | 38 |

are most effective in cases where the judiciary exercises greater discretion, whereas they are ineffective in cases involving third-party public power intervention. This aligns with economic intuition and practical observations.

## (ii) Charitable donations, geographical distribution and litigation cycle

The impact of charitable donations on the litigation cycle may be influenced by regional variations. Consequently, we categorize the provinces housing listed companies into four regions: eastern, central, western, and northeastern, based on the economic zone classification on the National Bureau of Statistics of China's official website. This categorization aims to delve into the association between donation practices and litigation cycles across different geographical regions. Table 9 presents the results of the grouped regressions, where the odd-numbered columns reflect the results based on the full sample group, and the even-numbered columns represent the results based on the subset of listed companies acting as plaintiffs. If the coefficient of the donation variable is significantly negative, it indicates that judicial departments in that region are more likely to be positively influenced by the reputation benefits brought by donations, thereby shortening the litigation cycle. On the other hand, even if a listed company makes charitable donations, it is unlikely to significantly shorten the litigation cycle due to limited judicial resources. According to the regression results in the odd-numbered columns, listed companies in the Northeast region exhibit the most pronounced reduction in litigation cycles as a result of charitable donations. From the regression results in the even-numbered columns, when listed companies in the Eastern, Central, and Northeast regions act as plaintiffs, their charitable donations have a more significant negative impact on the litigation cycle,

**Table 9. Charitable donations, geographical distribution and litigation cycle.**

| Variables | (1) | (2) | (3) | (4) | (5) | (6) | (7) | (8) |
|---|---|---|---|---|---|---|---|---|
| | East | | Central | | West | | Northeast | |
| | Full Sample | Plaintiff | Full Sample | Plaintiff | Full Sample | Plaintiff | Full Sample | Plaintiff |
| D_donate | -0.065 | -0.140* | -0.125 | -0.513** | 0.318** | 0.289 | -0.717** | -0.300*** |
| | (0.077) | (0.085) | (0.279) | (0.223) | (0.128) | (0.190) | (0.270) | (0.0140) |
| Constant | -2.580*** | -4.041*** | -2.912** | -3.182 | -1.090 | -3.917** | 38.03*** | -2.764*** |
| | (0.798) | (0.837) | (1.454) | (2.490) | (1.252) | (1.705) | (4.517) | (0.102) |
| Controls | YES | YES | YES | YES | YES | YES | YES | YES |
| Year | YES | YES | YES | YES | YES | YES | YES | YES |
| Observations | 1,128 | 705 | 214 | 118 | 294 | 158 | 69 | 24 |

while the effect is relatively weaker in the Western region. Similar patterns are observed when using corporate donation size (Lndonate) in the regressions. The empirical results suggest that charitable donations by listed companies have a particularly significant negative impact on the litigation cycle for companies in the Northeast and for listed companies in the Eastern and Central regions when acting as plaintiffs. These results imply that compared to the Northeast, Eastern, and Central regions, judicial resources in the Western region are relatively limited. As a result, even if local companies engage in active charitable donations, it is unlikely to shorten their litigation duration.

### (iii) Charitable giving, market-based indices and the litigation cycle

To comprehensively assess the extent of local government intervention in the market economy and the robustness of the legal and regulatory framework in a given locale, we utilized the marketization level of the enterprise's location as a proxy. Based on the median marketization index of the enterprise's location, we segregated the sample into two distinct groups: high and low marketization indexes. Subsequently, we conducted regression analyses, and the findings are summarized in Table 10. Our empirical results reveal a pronounced negative correlation between the charitable donation practices of listed companies and the reduction in their litigation cycle in regions with lower marketization indexes. Specifically, charitable donations effectively shortened the litigation cycle in these regions. However, in regions with higher marketization indexes, this relationship was not significantly observable, suggesting that the negative association between charitable donations and litigation cycle diminishes with increasing marketization. This finding underscores the influence of the marketization process on the nexus between corporate philanthropy and litigation duration among listed firms. In regions with a lower level of marketization, government intervention and legal regulation tend to be comparatively robust. Consequently, charitable donations, as a positive corporate practice, are more likely to garner recognition from governmental and judicial institutions, potentially resulting in expedited litigation processes. However, as marketization advances, government involvement wanes, and the legal regulatory framework becomes increasingly intricate. This evolution diminishes the significance of charitable donations in litigation, rendering it challenging for listed companies to secure substantial litigation benefits through this means.

### Remainder

Charitable donations serve as a significant means for listed companies to establish and sustain their corporate reputation. Prior analyses have demonstrated that upholding corporate

**Table 10. Charitable giving, market-based indices and the litigation cycle.**

| Variables | (1) | (2) | (3) | (4) |
|---|---|---|---|---|
| | HIGH | | LOW | |
| D_donate | -0.065 | | -0.140 | |
| | (0.097) | | (0.085) | |
| Lndonate | | -0.004 | | -0.013** |
| | | (0.008) | | (0.006) |
| Constant | -2.232** | -2.296** | -1.571** | -1.774** |
| | (1.098) | (1.140) | (0.704) | (0.727) |
| Controls | YES | YES | YES | YES |
| Year | YES | YES | YES | YES |
| Observations | 830 | 830 | 880 | 880 |

reputation through charitable donations positively impacts companies embroiled in litigation. Drawing from the theory of reputation insurance, Fu Chao et al. [8] corroborated that litigation risks faced by Chinese listed companies significantly elevate the level of charitable donations in subsequent periods. This finding underscores the "reputational compensation" function of charitable donations, rather than solely its "reputational insurance" role. The concept of "reputational insurance" inherently holds a preventative aspect, implying that corporations engage in charitable donations to mitigate potential litigation risks. Consequently, to comprehensively evaluate the "reputational insurance" function of charitable donations, a deeper exploration of the relationship between current-period donations and potential litigation risks is warranted. To provide compelling evidence on whether corporate charitable donations serve as a "reputation insurance" or a "compensatory" means, we examine the correlation between current charitable donations and current litigation risk. If a significant negative correlation is observed, this would indicate that charitable donations effectively mitigate litigation risk, thus supporting the "reputation insurance" perspective. Conversely, if the relationship is not significant, it would suggest that charitable giving is primarily employed as a remedial strategy to repair or preserve corporate reputations that have been damaged by litigation, rather than serving as a proactive "reputation insurance" mechanism.

To thoroughly investigate the association between charitable donations and litigation risk, defendant risk, as well as the corresponding number of lawsuits and defendants among listed companies, this study constructs a panel dataset spanning from 2008 to 2021, utilizing litigation data and announcements of listed companies in year t. A panel model is employed to estimate the relationship between charitable donations and litigation risk, as well as defendant risk. Given that the explanatory variables, litigation involvement risk (Litigation) and defendant risk (Defendant), are dummy variables, a panel probit model is chosen to comprehensively analyze the full sample data. For count-type variables, such as the number of lawsuits (N_litigation) and the number of defendants (N_defendant), a panel Poisson model is utilized for regression analysis to ensure the precision and reliability of the results. Furthermore, to enhance the robustness of the findings, we include control variables that are strongly correlated with firms' litigation risk, including financial leverage, main business profit percentage, two-rights separation degree, equity concentration index 1, net profitability of total assets, growth capacity, audit unit, dual-job situation, cash holdings, and firm size. Additionally, we control for the logarithm of provincial GDP, year fixed effects, and industry fixed effects.

The regression results presented in Table 11 demonstrate that, at a 5% significance level, charitable donations by listed companies significantly mitigate the risk of litigation and the risk of being defendants. However, they do not notably reduce the total number of lawsuits or defendants. This empirical evidence suggests that charitable donations play a positive role in "reputation insurance" in the context of judicial litigation, significantly reducing the likelihood of corporate litigation. Nonetheless, the impact of charitable donations on diminishing the subsequent number of lawsuits is more constrained. The findings underscore that a favorable corporate reputation can lower the probability of involvement in lawsuits and becoming a defendant, thereby providing enterprises with some leeway in legal disputes. However, to effectively reduce the specific number of lawsuits, companies must adopt a multifaceted approach encompassing the enhancement of internal management, refinement of business strategies, and other pertinent efforts.

## Conclusions and implications of the study

This study, based on the litigation and arbitration data of A-share listed companies in Shanghai and Shenzhen from 2008 to 2021, provides an in-depth analysis of the motivations and

**Table 11. Charitable giving and potential litigation risks.**

| Variables | (1) | (2) | (3) | (4) | (5) | (6) | (7) | (8) |
|---|---|---|---|---|---|---|---|---|
| | xtprobit | xtprobit | xtprobit | xtprobit | xtpoison | xtpoison | xtpoison | xtpoison |
| | Litigation | Litigation | Defendant | Defendant | N_litigation | N_litigation | N_defendant | N_defendant |
| D_donate | -0.078*** | | -0.099** | | -0.010 | | -0.031 | |
| | (0.023) | | (0.040) | | (0.076) | | (0.086) | |
| Lndonate | | -0.006*** | | -0.008** | | -0.006 | | -0.008 |
| | | (0.002) | | (0.003) | | (0.005) | | (0.006) |
| Constant | 0.925** | 0.860** | 1.288** | 1.219** | 4.563*** | 4.341*** | 4.642*** | 4.389*** |
| | (0.403) | (0.405) | (0.556) | (0.562) | (1.173) | (1.161) | (1.595) | (1.582) |
| Controls | Year | Year | Year | Year | Year | Year | Year | Year |
| Year | Year | Year | Year | Year | Year | Year | Year | Year |
| Industry | Year | Year | Year | Year | Year | Year | Year | Year |
| Observations | 37,912 | 37,911 | 9,102 | 9,102 | 9,109 | 9,109 | 9,109 | 9,109 |
| Number of scode | 4,234 | 4,234 | 2,434 | 2,434 | 2,436 | 2,436 | 2,436 | 2,436 |

effects of charitable donations by Chinese listed companies in the context of judicial litigation. The study finds that charitable donations by listed companies effectively shorten their litigation cycle (the time interval from the filing to the resolution of a case). This effect is particularly pronounced when the listed company acts as the plaintiff, but relatively insignificant when the company is the defendant. Furthermore, we further verify that the negative relationship between charitable donations and the litigation cycle is mediated through a reputation mechanism. Charitable donations significantly enhance the company's reputation rating, which in turn brings potential litigation benefits, thereby reducing the litigation cycle. Third, heterogeneity analysis reveals the differentiated effects of charitable donations across different types of cases and regions. Compared with administrative and criminal cases, charitable donations have a more significant effect on shortening the litigation cycle when listed companies face civil and arbitration cases. Moreover, in the eastern, central, and northeastern regions, charitable donations have a more significant negative impact on the litigation cycle when the listed company is the plaintiff, while the effect is weaker in the western regions. In regions with lower levels of marketization, the negative impact of charitable donations on reducing the litigation cycle is more pronounced. Finally, this paper attempts to respond to the study by Fu Chao et al. [8] by constructing a panel dataset of corporate litigation and whether a company is the defendant from 2008 to 2021. The empirical results show that corporate charitable donations not only have a reputation "compensating" function but also act as a reputation "insurance." The study reveals that when faced with judicial litigation, listed companies tend to reduce litigation time through legal means such as charitable donations in order to lower high judicial costs and maintain their corporate reputation. Through charitable donations, companies can not only repair damaged reputations but also build a reputation safeguard for themselves in potential litigations, reducing litigation risks and ensuring the long-term stable development of the company.

Based on the conclusions of this study, we attempt to offer policy recommendations from both the corporate and judicial perspectives to promote judicial system reform and the achievement of common prosperity:

1. Listed companies should establish a linkage mechanism between charitable donations and reputation management, enhancing the integration of charitable donations with corporate strategy. By leveraging donations to improve the company's reputation rating, companies

can utilize this reputational advantage in litigation, enhancing their bargaining power in lawsuits and subsequently reducing litigation costs and duration.

2. Promote judicial system reform and innovation, strengthening judicial transparency and credibility. Judicial departments should actively explore the establishment of a more efficient and fair litigation system, and strengthen the supervision and evaluation of the relationship between charitable donations and litigation preferences, preventing abuse and corruption.

3. Strengthen the promotion of the rule of law and legal education, facilitating the balanced distribution of judicial resources across regions. Judicial departments should appropriately increase support for the central and western regions to enhance their judicial capacity and efficiency. Meanwhile, companies in the eastern regions should be encouraged to support the judicial construction and development of the central and western regions through charitable donations and other means.

This study expands the research on the motivations behind charitable donations by listed companies into the realm of judicial litigation; however, there are still some limitations in the study. First, due to data limitations, panel data could not be used in the benchmark regressions, which restricts the depth of the analysis. Second, in addition to the reputation mechanism analyzed in this paper, political connections and social networks may also serve as potential mechanisms influencing the litigation cycle of listed companies. Due to space constraints, these aspects could not be explored in this paper. Future research can build upon the resolution of these limitations and conduct more detailed and in-depth studies.

## Supporting information

**S1 Appendix. The statutory time limit for first-instance trials in civil, administrative, criminal, and arbitration cases in China.**
(DOCX)

## Author Contributions

**Data curation:** Feng Zhu.

**Formal analysis:** Feng Zhu.

**Visualization:** Feng Zhu.

**Writing – original draft:** Feng Zhu.

**Writing – review & editing:** Feng Zhu.

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
