## [Decision Letter · Decision Letter 0]

15 Oct 2024

PONE-D-24-30529How Charitable Giving Affects Litigation Efficiency? Empirical Evidence from ChinaPLOS ONE

Dear Dr. ZHU,

Thank you for submitting your manuscript to PLOS ONE. After careful consideration, we feel that it has merit but does not fully meet PLOS ONE’s publication criteria as it currently stands. Therefore, we invite you to submit a revised version of the manuscript that addresses the points raised during the review process.

**Reviewers 1, 2, and 4 provided positive comments. During this revision, please ensure that you thoroughly address all of the reviewers’ recommendations. If you have a different interpretation of any of the reviewers’ comments, please provide a detailed explanation of your reasoning.**

**I would also like to highlight a specific concern raised by Reviewer 3 regarding the potential for the same or a similar manuscript to be published in two languages, namely English and Chinese. This situation could present copyright issues related to your manuscript. Please submit a detailed response to this matter.**

We look forward to receiving your revised manuscript.

Kind regards,

Jianpeng Fan

Academic Editor

PLOS ONE

**Journal Requirements:**

2. Please note that your Data Availability Statement is currently missing the DOI/accession number of each dataset or a direct link to access each database. If your manuscript is accepted for publication, you will be asked to provide these details on a very short timeline. We therefore suggest that you provide this information now, though we will not hold up the peer review process if you are unable.

Reviewers' comments:

Reviewer's Responses to Questions

**Comments to the Author**

1. Is the manuscript technically sound, and do the data support the conclusions?

Reviewer #1: Yes

Reviewer #2: Partly

Reviewer #3: Yes

Reviewer #4: Yes

2. Has the statistical analysis been performed appropriately and rigorously? 

Reviewer #1: Yes

Reviewer #2: No

Reviewer #3: Yes

Reviewer #4: Yes

3. Have the authors made all data underlying the findings in their manuscript fully available?

Reviewer #1: Yes

Reviewer #2: Yes

Reviewer #3: Yes

Reviewer #4: No

4. Is the manuscript presented in an intelligible fashion and written in standard English?

Reviewer #1: Yes

Reviewer #2: Yes

Reviewer #3: Yes

Reviewer #4: Yes

5. Review Comments to the Author

**Reviewer #1: **PONE-D-24-30529

Title: How Charitable Giving Affects Litigation Efficiency? Empirical Evidence from China

The research article is well-written and demonstrates thorough research. The research methodology is organized, and the data collection method is impressive. However, there are still parts that need correction and improvement. Please follow the comments and suggestions below.

1. The author provides a literature review as a very short communication, so it is strongly recommended that a more specific literature review be provided, which would be beneficial for the reader.

2. Could the author please explain more about the functions of Articles 164 and 152 of the Code of Civil Procedure and what main role they play in governing litigation and the legal system in this study?"

3. The conclusion is too lengthy and resembles an introduction. Please revise and rewrite it clearly. Additionally, I suggest that the author create a separate section for the conclusion, limitations, and directions for future research.

4. Please, the author, carefully revise the paragraph by paragraph; I see some references are missing, so please provide the citation below and also carefully check others in your manuscript.

Corporate reputation represents a significant intangible asset that profoundly influences consumer behavior, market direction, and societal opinion. The events surrounding Wahaha in 2024 underscore the substantial impact corporate reputation can have on stakeholders such as consumers, competitors, and investors. Consequently, numerous enterprises, particularly listed companies, are augmenting their investments in reputation assets, with charitable donations serving as a pivotal strategy

The marginal contributions of this paper are fourfold. Firstly, it broadens the research scope of corporate charitable donations to the realm of judicial litigation, empirically examining the actual impact of such donations on the litigation cycle of listed companies.

In response to the first logical challenge, we employ Reputation Theory to elucidate the underlying mechanisms.

Reputation Theory, originating in the 19th century with economists such as Adam Smith, underscores the significance of reputation as a comprehensive public evaluation of an individual or organization's past behavior and performance in a specific social context.

I am very happy to review this scientific research, and I look forward to the revisions. I believe the author will make the necessary improvements. Good Luck!

**Reviewer #2:** This manuscript examines the impact of charitable donations on litigation efficiency. The theoretical discussion of this study is relatively standard, but the flaw lies in its empirical evidence.

There are several issues in the econometric analysis:

(1) The regression model does not control for any fixed effects.

(2) It is not clear at what level clustered standard errors are controlled for.

(3) There is a lack of justification for the selection of IV, particularly whether it meets the requirements of relevance and exogeneity.

(4) The mechanism analysis only examines the impact of charitable donations on corporate reputation, but lacks justification for the complete causal chain of “charitable donations → corporate reputation → litigation efficiency.” It is also reasonable to question whether there might be a competing causal relationship of “charitable donations → litigation efficiency → corporate reputation.”

(5) Table 11 is insufficient to support the examination of the "reputation insurance" function of charitable donations, as the argument is at least incomplete. It is not detailed whether litigation involvement risk and defendant risk differentiate between the current and lagged periods.

(6) In Table 4, the two groups have observations of 1010 and 633, but the sum does not equal the total of 1710.

(7) In columns 2 and 4 of Table 4, the regression coefficients for law and government are insignificant, which contradicts the discussion in the main text.

(8) Heterogeneity analysis shows that in the western regions of China, the regression coefficient for charitable donations is significantly positive, and there is no reasonable explanation for this result.

The biggest challenge of this manuscript lies in the insufficient empirical evidence.

**Reviewer #3:** Please note that this article, which is very structured and persuasive, should not be published twice, particularly in journals from China, if you agree to be accepted.

Though using the term "efficiency" in the title, I see no strong connection in the main body, where relates more to "litigation duration". Think about changing the term in the main body, or the title. I prefer to "duration", since efficiency is a much complicated term.

Though the data is available in the web, you may probably tell me the link of the data source in the footnote, so I can be more convinced. I have searched two websites you provided. There are many data within, and I can't identify which data you are using.

Apart from what mentioned above, I think it is a valuable article.

**Reviewer #4:** This is an intriguing study that elucidates how charitable donations, by influencing corporate reputation, alter case processing order and judicial resource allocation, thereby providing a novel theoretical foundation for understanding the complexity and externalities of judicial activities and enriching legal and economic research. The study employs quantitative methods and adequately explains its methodology and data, leading to highly credible conclusions. However, there is several areas for improvement:

1.While the paper addresses endogenous issues using instrumental variables, it would be beneficial to delve into other possible sources, including reverse causality or omitted variable bias.

2.The core contribution is the reputation mechanism as a pathway for charitable donations to impact litigation duration, but its exact function needs deeper revelation. Intermediate variable analysis is suggested to clarify how donations affect litigation through reputation. Additionally, other potential mechanisms like political connections and social networks should not be overlooked; their relationships with variables should be addressed even if not expanded upon.

3.The paper explores the influence of case types and regions on the donation-litigation relationship but lacks discussion on the theoretical mechanisms behind these heterogeneous effects. Its analysis of regional variations in donation, allocation, and litigation duration is somewhat cursory.

4.The mention of the Civil Code's maximum trial duration for civil suits on page 4 contrasts with later comparisons involving administrative, criminal, and arbitration procedures for listed companies. China's legislation sets different trial duration rules for various procedures, which should be distinguished.

5.The conclusion and implications section should further propose specific policy recommendations, such as encouraging corporate donations and improving the judicial system for litigation efficiency, particularly addressing whether current trial duration rules need revision.

6. PLOS authors have the option to publish the peer review history of their article (what does this mean?). If published, this will include your full peer review and any attached files.

Reviewer #1: No

Reviewer #2: No

Reviewer #3: No

Reviewer #4: No

---

## [Author Response · Author response to Decision Letter 0]

28 Nov 2024

Dear Reviewers,

I would like to express our sincere gratitude to the four reviewers for their careful review and valuable suggestions. I am honoured and encouraged to receive your review comments. Your professional scrutiny, profound insights, and insistence on academic rigour not only provided valuable guidance for our research, but also prompted me to constantly reflect and improve, and strive to make the paper even better.

I have carefully reviewed the comments and suggestions from four reviewers, and have made the necessary revisions to address each point. I believe that these changes have strengthened the paper's theoretical framework, methodological approach, and empirical findings. Below is a summary of the revisions I have made.

 Reviewer-1

 Observations Reply indexing

1 The author provides a literature review as a very short communication, so it is strongly recommended that a more specific literature review be provided, which would be beneficial for the reader. In accordance with your valuable suggestion, I have provided a more specific literature review in the main body of the paper. This review offers a more detailed presentation of the research findings related to charitable giving, litigation duration, and the interrelationship between these two aspects. Based on this expanded review, I have highlighted the significance and novelty of the current study. Specifically, I have organized the literature into two main sections:

(1)Research on the Motivations for Charitable Giving and CSR:

Through a systematic review and analysis of the literature, it becomes evident that existing research on the relationship between charitable giving and judicial activities primarily focuses on the early litigation risks and later litigation outcomes. However, it overlooks an in-depth examination of the litigation process itself.

(2)Research on External Factors Influencing Judicial Decision-Making:

The analysis reveals that Chinese judicial bodies, when handling cases involving listed companies, may be influenced by factors beyond the law (such as corporate charitable donations). This influence appears to affect the litigation cycle at the procedural law level. See pages 6-11 of the Manuscript.

2 Could the author please explain more about the functions of Articles 164 and 152 of the Code of Civil Procedure and what main role they play in governing litigation and the legal system in this study?"

 (1)On the Functions of Article 164 and Article 152 of the Civil Procedure Law:

Articles 164 and 152 of the Civil Procedure Law stipulate the timeframes for the trial of civil cases under the simplified procedure and the ordinary procedure, respectively. For cases subject to the simplified procedure and ordinary procedure, the time from filing to conclusion is set as "3+1+x" and "6+6+x," meaning that, under normal circumstances, a civil lawsuit should be concluded within 4 months and 12 months, respectively.

(2)On the Role of Articles 164 and 152 of the Civil Procedure Law:

The inclusion of these articles is intended to demonstrate that the duration of litigation can be assessed and balanced by the court or the presiding judge. Naturally, such assessments are subject to various external factors that may influence this balancing process, such as the positive reputation effects generated by charitable donations, as discussed in this paper. This also reflects the theoretical exploration of a causal relationship between charitable giving and litigation duration. 

3 The conclusion is too lengthy and resembles an introduction. Please revise and rewrite it clearly. Additionally, I suggest that the author create a separate section for the conclusion, limitations, and directions for future research. In accordance with your suggestion, I have streamlined the conclusion section and reorganized it following a logical structure: conclusion, recommendations, limitations, and directions for future research. See pages 36-38 of the Manuscript.

4 Please, the author, carefully revise the paragraph by paragraph; I see some references are missing, so please provide the citation below and also carefully check others in your manuscript. I have added the references that you pointed out as missing in the paper and have thoroughly reviewed and refined the other sections of the manuscript. Below are the relevant paragraphs and their corresponding reference sources as mentioned.

(1)Corporate reputation represents a significant intangible asset that profoundly influences consumer behavior, market direction, and societal opinion(Guan et al., 2019). [ Guan Kaolei, Zhang Rui. Corporate Reputation and Earnings Management: A Perspective of Effective Contracts or Rent-Seeking? [J]. Accounting Research, 2019, (01): 59-64(in Chinese).]The events surrounding Wahaha in 2024 underscore the substantial impact corporate reputation can have on stakeholders such as consumers, competitors, and investors. Consequently, numerous enterprises, particularly listed companies, are augmenting their investments in reputation assets, with charitable donations serving as a pivotal strategy(Gao et al., 2012).[ Gao Yongqiang, Chen Yajing, Zhang Yunjun. Corporate Reputation, Charitable Donations, and Consumer Responses [J]. Contemporary Economic Management, 2012, 34(06): 20-25(in Chinese).]

(2)The marginal contributions of this paper are fourfold. Firstly, it broadens the research scope of corporate charitable donations to the realm of judicial litigation, empirically examining the actual impact of such donations on the litigation cycle of listed companies(Fu et al., 2017).[ Fu Chao, Ji Li. Litigation Risk and Corporate Charitable Donations: An Explanation from the Perspective of "Reputation Insurance" [J]. Nankai Business Review, 2017, 20(02): 108-1211 (in Chinese).]

(3) In response to the first logical challenge, we employ Reputation Theory to elucidate the underlying mechanisms(Jean, 1996).[ Tirole, Jean. “A Theory of Collective Reputations (with applications to the persistence of corruption and to firm quality).” The Review of Economic Studies 63 (1996): 1-22.]

(4) Reputation Theory, originating in the 19th century with economists such as Adam Smith, underscores the significance of reputation as a comprehensive public evaluation of an individual or organization's past behavior and performance in a specific social context (Smith, 1763). 

 Reviewer-2

1 The regression model does not control for any fixed effects.

 I have included controls for provincial and year fixed effects in the regression model. To enhance readability for the readers, I have strengthened the corresponding explanations in the model and variable settings section. See page of 14-15the Manuscript.

2 It is not clear at what level clustered standard errors are controlled for. The standard errors used in this paper are heteroscedasticity-robust standard errors. Since the data employed in this study consists of mixed cross-sectional data, which includes multiple distinct individuals, heteroscedasticity is commonly present. Therefore, heteroscedasticity-robust standard errors are applied. Additionally, the type of standard errors used is indicated in the manuscript. See footnote to table 4 on page 21 of the Manuscript.

3 There is a lack of justification for the selection of IV, particularly whether it meets the requirements of relevance and exogeneity. The use of average donations by province, industry, and enterprise nature as an instrumental variable in this paper fulfills the criteria for relevance and exogeneity.Furthermore, the rationale for selecting the instrumental variables is further elaborated in the main text. See page 24-25 of the Manuscript.

4 The mechanism analysis only examines the impact of charitable donations on corporate reputation, but lacks justification for the complete causal chain of “charitable donations → corporate reputation → litigation efficiency.” It is also reasonable to question whether there might be a competing causal relationship of “charitable donations → litigation efficiency → corporate reputation. In response to your suggestion, I have refined the full causal chain argument of "charitable donations → corporate reputation → litigation efficiency." Specifically, I employed the Poisson regression method to capture the mapping relationship between charitable donations and reputation. Additionally, I used the same ordinary least squares (OLS) model as in the baseline regression to fit the relationship between corporate reputation and litigation duration (the supplementary regression results are presented in columns 7 and 8 of Table 7). The refined regression results support the proposed "charitable donations → corporate reputation → litigation efficiency" mechanism and reject the competing causal relationship of "charitable donations → litigation efficiency → corporate reputation." See pages 26-27 of the Manuscript.

5 Table 11 is insufficient to support the examination of the "reputation insurance" function of charitable donations, as the argument is at least incomplete. It is not detailed whether litigation involvement risk and defendant risk differentiate between the current and lagged periods. (1)On the Argument for Charitable Donations as "Reputation Insurance"

I used the charitable donations of listed companies in the current year to replace the donations in the subsequent period (Fu et al., 2017), which largely supports the "reputation insurance" function of charitable donations in contrast to existing studies. This is because corporate charitable donations are made continuously throughout the year, and according to descriptive statistics, the average litigation duration is one year. Thus, the charitable donations made by firms in the current year effectively precede the conclusion of the litigation.

(2)Explanation of Litigation Risk and Defendant Risk in the Current and Lagged Periods

There are two reasons for directly using the current period's donations and litigation risk: First, I referred to studies by Wang et al. (2008) and Lin et al. (2015), and found that in the study of litigation risk, there is limited consideration of lagged effects in academic research. Second, as explained in the first point, since the average litigation duration is one year, corporate charitable donations in the current year precede the ongoing litigation in that same period. Therefore, there is no need to apply a lagged treatment. 

6 In Table 4, the two groups have observations of 1010 and 633, but the sum does not equal the total of 1710. The reason why the number of observations in the grouped regression (1010 + 633 = 1643) is smaller than the total sample size of 1710 is that, in the variable representing party identity, in addition to the "plaintiff" and "defendant" categories, there are a small number of observations classified as "third party" or "unspecified." To enhance clarity for readers, I have included a footnote in Table 4 to provide an explanation. See page 21 of the Manuscript.

7 In columns 2 and 4 of Table 4, the regression coefficients for law and government are insignificant, which contradicts the discussion in the main text. The regression coefficients for law and government presented in the main text are based on columns (5) and (6) of Table 4. To avoid potential misunderstandings, I have revised the corresponding statements in the manuscript accordingly. See pages 21-22 of the Manuscript.

8 Heterogeneity analysis shows that in the western regions of China, the regression coefficient for charitable donations is significantly positive, and there is no reasonable explanation for this result. In the subsample analysis for the western region, the significantly positive regression coefficient of charitable donations may be attributed to the disproportionate number of cases where firms from this region are defendants. To address this, I conducted a regional heterogeneity analysis focusing on subsamples where listed companies act as plaintiffs and incorporated the corresponding regression results and interpretations into the manuscript. The results indicate that charitable donations by listed companies in the northeastern region exhibit the most significant effect in reducing litigation duration. Furthermore, when firms in the eastern, central, and northeastern regions serve as plaintiffs, the negative impact of charitable donations on litigation duration is more pronounced, whereas this effect is weaker in the western region. See pages 29-31 of the Manuscript.

 Reviewer-3

1 Please note that this article, which is very structured and persuasive, should not be published twice, particularly in journals from China, if you agree to be accepted.

Though using the term "efficiency" in the title, I see no strong connection in the main body, where relates more to "litigation duration". Think about changing the term in the main body, or the title. I prefer to "duration", since efficiency is a much complicated term.

Though the data is available in the web, you may probably tell me the link of the data source in the footnote, so I can be more convinced. I have searched two websites you provided. There are many data within, and I can't identify which data you are using.

Apart from what mentioned above, I think it is a valuable article. Thank you for your recognition of this study and for your valuable feedback. Based on your comments, I have made the following responses and revisions:

(1)On the Appropriateness of the Term "Efficiency" in the Title

As you pointed out, the main focus of the study lies in "litigation duration" rather than "efficiency." To avoid any potential misinterpretation, I have revised the title to replace "efficiency" with "duration." Additionally, all relevant terms in the main text have been updated to ensure consistency with the revised title.

(2)On the Clarity of Data Sources

The data that support the findings of this study are openly available in Data and procedures at http://dx.doi.org/10.5281/zenodo.14238241

If you need the original data you can find it by using the following database addresses and filters:

Charitable Giving Data: This was obtained from the CSMAR database. After accessing the database, search using the Chinese terms “non-operating income” and “non-operating expenditure.” After retrieving the raw data, manual sorting of donation items is required to calculate the annual donation amounts for listed companies.

Litigation Data: This was retrieved from the CNRDS database. Use the Chinese term “Listed Company Litigation and Arbitration Statistics” in the search bar to locate the relevant data.

Control Variable Data: This was sourced from the CSMAR database. Simply search for the names of the specific control variables in the search bar to retrieve the data.

If you have any questions regarding the data, please feel free to contact me. I am committed to ensuring the accuracy, transparency, and traceability of the data used in this study, as I recognize the importance of data validation in the review process.

(3)On Concerns About Duplicate Submissions

Thank you for raising this critical point. I assure you that this study is entirely original and has not been published in any other journal nor submitted elsewhere. I will maintain close communication with the editorial office to ensure full compliance with academic ethics throughout the publication process. Furthermore, I solemnly commit that if this article is accepted , it will not be published or disseminated in any form in other journals, including those in China. I strictly adhere to principles of academic integrity and will ensure the uniqueness and exclusivity of this article.

Thank you again for your detailed feedback and support. Please do not hesitate to reach out if further clarification or information is required. 

 Reviewer-4

1 While the paper addresses endogenous issues using instrumental variables, it would be beneficial to delve into other possible sources, including reverse causality or omitted variable bias. I have addressed the potential endogeneity issues arising from reverse causality and omitted variable bias in the manuscript. To mitigate these concerns, I employed an instrumental variable (IV) approach to reduce estimation errors caused by potential endogeneity. The results obtained using the IV me

---

## [Decision Letter · Decision Letter 1]

9 Dec 2024

How charitable giving affects litigation duration?

Empirical evidence from China

PONE-D-24-30529R1

Dear Dr. ZHU,

We’re pleased to inform you that your manuscript has been judged scientifically suitable for publication and will be formally accepted for publication once it meets all outstanding technical requirements.

Kind regards,

Jianpeng Fan

Academic Editor

PLOS ONE

Additional Editor Comments (optional):

Thank you for your efforts. At present, most reviewers have recommended accepting the manuscript, while one reviewer suggests adding a few related references. Here, I would like to remind you to carefully verify the relevance of these recommended references to the manuscript and decide accordingly whether to accept such suggestions.

Reviewers' comments:

Reviewer's Responses to Questions

**Comments to the Author**

1. If the authors have adequately addressed your comments raised in a previous round of review and you feel that this manuscript is now acceptable for publication, you may indicate that here to bypass the “Comments to the Author” section, enter your conflict of interest statement in the “Confidential to Editor” section, and submit your "Accept" recommendation.

Reviewer #1: All comments have been addressed

Reviewer #2: All comments have been addressed

Reviewer #4: All comments have been addressed

2. Is the manuscript technically sound, and do the data support the conclusions?

Reviewer #1: Yes

Reviewer #2: Yes

Reviewer #4: Yes

3. Has the statistical analysis been performed appropriately and rigorously? 

Reviewer #1: Yes

Reviewer #2: Yes

Reviewer #4: I Don't Know

4. Have the authors made all data underlying the findings in their manuscript fully available?

Reviewer #1: Yes

Reviewer #2: Yes

Reviewer #4: Yes

5. Is the manuscript presented in an intelligible fashion and written in standard English?

Reviewer #1: Yes

Reviewer #2: Yes

Reviewer #4: Yes

6. Review Comments to the Author

Reviewer #1: Thank you for the author's revision, which included following the reviewers’ suggestions and demonstrating good data management in the manuscript. I strongly recommend that the manuscript be published after minor corrections. Please, author, add a section “Future Research Directions” after the Conclusions section. Additionally, I suggest citing the newly published research articles related to the author’s study, which I have provided below.

The requirements for filing a consumer public interest litigation in China.

https://doi.org/10.1504/MEJM.2024.135152

Digital Technology Application and Enterprise Competitiveness: The Mediating Role of ESG Performance and Green Technology Innovation. Environ Dev Sustain 2023, doi:10.1007/s10668-023-03979-3.

The Role of Technology in the Digital Economy’s Sustainable Development of Hainan Free Trade Port and Genetic Testing: Cloud Computing and Digital Law. https://doi.org/10.3390/su16146025

Management Economic Systems and Governance to Reduce Potential Risks in Digital Silk Road Investments: Legal Cooperation between Hainan Free Trade Port and Ethiopia. https://doi.org/10.3390/systems12080305

Reviewer #2: I am very glad to find that the author has fully followed my suggestions and has made a lot of detailed revisions, which effectively solve technical problems in the previous round of review. I feel that this manuscript is now acceptable for publication.

Reviewer #4: I think the author really addressed my comments well after taking a close look at the manuscript. Therefore, I recommend publishing this article.

7. PLOS authors have the option to publish the peer review history of their article (what does this mean?). If published, this will include your full peer review and any attached files.

Reviewer #1: No

Reviewer #2: No

Reviewer #4: No

---

## [Editor Report · Acceptance letter]

13 Dec 2024

PONE-D-24-30529R1 

PLOS ONE

Dear Dr. Zhu, 

I'm pleased to inform you that your manuscript has been deemed suitable for publication in PLOS ONE. Congratulations! Your manuscript is now being handed over to our production team.

Kind regards, 

on behalf of

Dr. Jianpeng Fan 

Academic Editor

PLOS ONE